# Neuroenhancement in French and Romanian University Students, Motivations and Associated Factors

**DOI:** 10.3390/ijerph18083880

**Published:** 2021-04-07

**Authors:** Irina Brumboiu, Alessandro Porrovecchio, Thierry Peze, Remy Hurdiel, Irina Cazacu, Cristina Mogosan, Joel Ladner, Marie-Pierre Tavolacci

**Affiliations:** 1Cluj-Napoca Unit—The Net-Work of International Francophone Clinical Epidemiology, Iuliu Hatieganu University of Medicine and Pharmacy, 400000 Cluj-Napoca, Romania; irinabrumboiu@gmail.com (I.B.); cazacuirina@yahoo.com (I.C.); cmogosan@umfcluj.ro (C.M.); 2Univ. Littoral Côte d’Opale, Univ. Lille Artois ULR 7369—URePSSS—Unité de Recherche Pluridiscipli-Naire Sport Santé Société, F-59375 Dunkerque, France; alessandro.porrovecchio@gmail.com (A.P.); thierry.peze@univ-littoral.fr (T.P.); remy.hurdiel@univ-littoral.fr (R.H.); 3Clinical Investigation Center 1404 and INSERM 1073, Rouen University Hospital and Rouen Normandie University, 76000 Rouen, France; joel.ladner@chu-rouen.fr; 4Department of Epidemiology and Health Promotion and INSERM 1073, Rouen University Hospital and Rouen Normandie University, 76000 Rouen, France

**Keywords:** neuroenhancement, prescription drugs, drugs of abuse, soft enhancers, students

## Abstract

This cross-sectional study aimed to determine the use of neuroenhancers, the motivations and factors associated with their use in French and Romanian university students. Students from two universities in France (Rouen and Opal Coast University) and one in Romania (Cluj-Napoca) were asked to complete a self-administered anonymous questionnaire, either online or on paper, about the use of three different categories of substance: Prescription drugs (methylphenidate, modafinil, and beta-blockers), drugs of abuse (alcohol, cannabis, cocaine, and amphetamines), and soft enhancers (coffee, vitamins, caffeine tablets, and energy drinks). In total, 1110 students were included: The users were 2.2% for prescription drugs, 4.3% for drugs of abuse, and 55.0% for soft enhancers. Students used neuroenhancement to stay awake for study (69.3%), to improve concentration (55.5%), to decrease stress (40.9%), and to improve memory (39.6%). Neuroenhancement was considered to meet expectations by 74.4% of users. The factors associated with the use of drugs of abuse were frequent binge drinking (Adjusted Odds Ratio—AOR: 6.49 [95% CI: 2.53–16.6]), smoking (AOR: 5.50 [95% CI: 2.98–10.14]), having a student job (AOR: 2.42 [95% CI 1.13–5.17]), and being male (AOR: 2.23 [95% CI:1.21–4.11]). No significant associations with eating disorders were detected for any of the three categories of substances. University students reported neuroenhancement with prescription drugs, drugs of abuse, and mainly soft enhancers. These substances were used mainly to increase the waking hours. Educational programs in universities seem to be required in order to increase student awareness of the problems caused by neuroenhancements, and to decrease the associated risks by changing students’ attitudes and beliefs.

## 1. Introduction

Neuroenhancement in healthy subjects is the improvement of cognitive, emotional and motivational functions through the consumption of various categories of substances [1]. Cognitive enhancers may be readily available substances, such as caffeine and energy drinks, or psychostimulants, when used to improve cognitive functions [2]. Caffeinated products, food supplements, and energy drinks are considered “soft enhancers”, whereas prescription drugs (beta-blockers, modafinil, and methylphenidate) and drugs of abuse (alcohol, cannabis, cocaine, and amphetamines) are defined as “neuroenhancers” [3]. University students are at high risk of neuroenhancement [4]. Some studies suggested that neuroenhancers were helpful for improving concentration, making it possible to study for longer [5], to increase working memory performance, to boost self-esteem, and to cope with stressors (academic overload, competition with peers, constant pressure to succeed, financial burden) [5,6]. This risky behavior appears to increase the likelihood of using the stimulant again, at higher doses and/or in combination with drugs.

There are numerous concerns regarding neuroenhancement in Europe [4,7,8,9,10,11]. A study of Swiss university students found that 13.8% had tried to enhance their cognitive performance at least once with prescription drugs (7.6%) or drugs of abuse (7.8%) [4]. In France, data relative to neuroenhancement among university students are scarce. In the English literature, only two studies were published. The first one, conducted in 2014, reported that among a non-representative sample of 206 medicine and pharmacology students questioned, 67.4% declared to have consumed at least one cognitive enhancer in the past year: The majority consumed vitamin C (84.8%) and caffeine tablets (71.9%), and only 5.8% used prescription drugs [12]. The second study was conducted in 2016. Lifetime prevalence of cognitive enhancer use among undergraduate and postgraduate medical students was 33%. The commonly used substances were caffeine tablets and energy drinks (29.7%), followed by prescription drugs (6.7%) and illicit drugs (5.2%) [10]. These studies investigated prevalence of neuroenhancement only among health students, while neglecting participants with other curriculums who are also susceptible to consume neuroenhancers. Furthermore, the investigated substances do not represent all categories of substances consumed for cognitive enhancement [10,12].

When comparing the different countries, the highest lifetime prevalence rates of alcohol and drug use were found among Eastern European countries [13,14]. This high prevalence of drugs and alcohol use in the general young population could influence their use exclusively for the purposes of cognitive enhancement in Romanian universities. To date, no studies have been performed to either confirm or refute this hypothesis.

There is no study investigating the prevalence of all categories of cognitive enhancers among French and Romanian students with different study curriculums. We conducted a cross-sectional study to determine the use, motivations, and factors associated with the use of soft enhancers, drugs of abuse, and prescription drugs among French and Romanian university students.

## 2. Materials and Methods

### 2.1. Study Design

A cross-sectional study was conducted among voluntary university students aged between 18 and 25 years from Cluj-Napoca University in Romania or from the Rouen University or from Opal Coast University in France. University students were invited by email to complete an online anonymous 20-min self-administered questionnaire (30,000 at Rouen University, 1600 at Opal Coast University, and 1100 in Cluj Napoca). Student participation was voluntary, leading to a convenience sample rather than representative. This study was approved by the Rouen University Hospital and Cluj-Napoca University Institutional Review Boards.

### 2.2. Data Collection

#### 2.2.1. Sociodemographic Characteristics

Data about age, sex, grant-holder status, student employment status, smoking and binge drinking (never, occasional: Once a month or less, and frequent: Twice a month or more) were collected.

#### 2.2.2. University Curriculum

Faculties and schools specializing in medicine, pharmacy, nursing, physiotherapy, midwifery, and radiology technologist studies were included in the “healthcare” group. All other specialties were included in the “mixed discipline university” group. The academic year of study was collected.

#### 2.2.3. Stress Evaluation

The Perceived Stress Scale (PSS) is the most widely used psychological instrument for measuring the perception of stress [15]. It assesses the perceived state of stress by allowing the subjects to estimate the extent to which they can control a situation. Each of the 10 items is scored from 0 to 4. Higher scores correspond to higher levels of perceived stress (linear relationship). The score measured in this way allowed to make comparisons between individuals.

Eating disorders (ED): The SCOFF questionnaire is a screening tool for identifying the risk of EDs, such as anorexia nervosa, bulimia nervosa, and other unspecified EDs in young adults. It gives a score from 0 to 5, according to the number of positive answers. This questionnaire has been shown to be a highly effective screening instrument, with excellent sensitivity and specificity for detecting EDs on the basis of at least two positive answers [16]. A positive SCOFF score indicates that there were at least two positive responses to the five “yes/no” questions. This questionnaire was translated into French and the translated version (SCOFF-F) was validated [17]. The same SCOFF free access self-reporting questionnaire was used in Cluj-Napoca university [18].

#### 2.2.4. Assessment of Substance Use for Neuroenhancement

The questions were related to the individual consumption of each psychoactive substance to improve individual performance during revisions or examinations, in a non-festive setting. Three categories of substances were defined. (1) Prescription drugs included beta-blockers, methylphenidate and modafinil. (2) Drugs of abuse included alcohol, cannabis, cocaine, and amphetamines, and (3) “Soft enhancers” included coffee, vitamins, caffeine tablets, and energy drinks. The possible answers were never, once, sometimes (unsystematic use for each examination but used more than once) or regularly (use for each examination). For prescription drugs and drugs of abuse, because of the low frequency of consumption, never was classified as no, and once, sometimes, or regularly was classified as yes. For soft enhancers, never or once was classified as no and sometimes or regularly as yes. The specific reasons for neuroenhancement use: Staying awake, improving concentration, improving memory, and reducing stress were reported and whether these reasons applied never, sometimes, often, or always. Students were also asked to indicate whether their substance use met the expectations (no, partially, fully, don’t know) and any symptoms experienced at least once after substance use (loss of appetite, anxiety, panic attack, aggressiveness, excessive sweating, sleep disorder, weariness, palpitation, and weight loss).

#### 2.2.5. Other Assessment of Substance Use

Individuals were considered to be regular smokers if they smoked at least one cigarette/day, and regular cannabis users if they used it at least 10 times/month [19]. Students who reported drinking five alcoholic drinks (four for female students) in a single sitting at least once per month were classified as regular binge drinkers, and those doing so but at a lower frequency (less than once per month) were considered to be occasional binge drinkers [20,21]. Cannabis use and binge drinking were recorded for the purpose of a recreational context.

### 2.3. Patient and Public Involvement

No students were involved in setting the research questions. Students were invited to participate in the study by each university. The main results will be displayed on the website www.tasanteenunclic.org (accessed on 3 March 2021).

### 2.4. Statistical Analysis

Categorical variables were described as percentages and 95% confidence interval (CI) and compared using Fisher’s exact test. Continuous variables were described by their mean and Standard Deviation (SD) and were compared using the Student’s t test. *p*-values < 0.05 were considered significant. A logistic regression was performed to evaluate the independent factors associated with the consumption of soft enhancers and drugs of abuse. Prescription drugs were not included in the multivariate analysis because of the small number of users (*n* = 24). Model 1 included sociodemographic variables (sex, age, university, student job, grant-holder status, curriculum, academic year), Model 2 was defined as model 1 + smoking and Model 3 as model 1 + binge drinking. Adjusted odds ratios (AORs) and their 95% CI were calculated.

## 3. Results

### 3.1. Baseline Characteristics of the Study Population

In total, 1110 students participated in the study: 534 from Rouen University, 358 from Opal Coast University and 218 from Cluj-Napoca University. The characteristics of the students at each university are reported in Table 1. The proportion of senior students (academic year of study > 3) was higher in the Cluj-Napoca University population, whereas first-year students were more frequent in the French university populations.

### 3.2. Neuroenhancement

Most respondents (55.0%; 95%CI (52.0–56.0)) reported the use of soft enhancers. Drugs of abuse (4.3%; 95%CI (3.1–5.7)) and prescription drugs (2.2%; 95%CI (1.4–3.3) were less used. The use of neuroenhancer over the last year is shown, by product and by university, in Table 2. Beta-blockers and methylphenidate were the most frequently used prescription drugs. Alcohol and cannabis were the most prevalent drugs of abuse, with stimulants, such as cocaine and amphetamines, much less frequently consumed for neuroenhancement purposes. For the substances in the soft enhancers category, 45.3% of the students used coffee, 26.9% used vitamins and 10% used energy drinks. Caffeine tablets were rarely used. Neuroenhancement was more prevalent among Romanian than French university students for all three categories of substances considered.

### 3.3. Factors Associated with Neuroenhancement

The characteristics of neuroenhancer users are summarized by substance category in Table 3. Prescription drug use was more prevalent among healthcare and senior students.

No significant differences in mean perceived stress score or eating disorders were detected between users and non-users of neuroenhancers, for any of the three categories.

After logistic regression, the factors associated with the consumption of drugs of abuse were frequent binge drinking (AOR = 6.49; 95%CI 2.53–16.60), smoking (AOR = 5.50; 95%CI 2.98–10.14), having a student job (AOR = 2.42; 95%CI 1.13–5.17), and being male (AOR = 2.23; 95%CI 1.21–4.11). The consumption of soft enhancers was associated with frequent (AOR = 3.34; 95%CI 1.88–4.13) or occasional binge drinking (AOR = 1.69; 95%CI 1.26–5.92), smoking (AOR = 2.71; 95%CI 1.94–3.80), healthcare curriculum (AOR = 1.38; 95%CI 1.01–1.87) (Table 4).

### 3.4. Reasons for Neuroenhancement, Side Effects and Satisfaction

The main motivations for using neuroenhancement are shown, by frequency (never, sometimes, often, and always) in Figure 1. The motivations for using a neuroenhancer were to stay awake while working (69.3%), to improve concentration (55.5%), to reduce stress (40.9%), and to improve memory (39.6%).

The most common adverse effects reported after the consumption of neuroenhancers were sleep disorders (16.4%), palpitation (11.9%), and weariness (7.9%) (Figure 2).

Three-quarters of the students reported that their neuroenhancement use met their expectations (partial for 58.5% and full for 15.9%), 40.5% of students declared an improvement (partial for 36.5% and full for 4%) in their academic performance, and one quarter did not know (Figure 3).

## 4. Discussion

This survey is the first to provide information about the use of a large range of products (prescription drugs, drugs of abuse, and soft enhancers), reasons for neuroenhancement, and the factors associated with neuroenhancement for university students at French and Romanian universities. The use in the previous year only for the neuroenhancement was 55.0% for soft enhancer consumption, 4.3% for use for drugs of abuse, and 2.2% for prescription drugs. These findings are consistent with the findings of previous European studies on this subject [3,21]). In our study, Romanian students had the highest rates of soft enhancer use (84.9%), and the trends observed were similar to those reported among students at Puerto Rico (88%), American (92%), and Omanian (97%) universities [22,23,24]. The most widely used substance in our cohort was coffee (45.3%), followed by vitamins (26.9%) and energy drinks (10%). Caffeinated beverages, such as coffee, tea, and energy drinks, are generally considered socially acceptable and form the basis of the coping strategies used by students to enhance cognitive function and to manage stressful academic situations [23,24,25]. However, the misuse of soft enhancers can have a number of adverse effects, including breathing problems, an abnormal heartbeat, increases in blood pressure, diuresis and natriuresis, a decrease in insulin sensitivity, high levels of irritability, and chronic daily headaches if the usual dose is not taken [24,25,26]. It might be important (for future studies) to ask about the dosing of these substances to learn more about patterns of use and any indications of risk.

We found a prevalence of drugs of abuse consumption for the neuroenhancement of 4.3% (8% among Romanians students and 3% among French students). The most frequently used were alcohol (4.0%) and cannabis (3.2%) followed by amphetamines (0.8%) and cocaine (0.4%). Lifetime prevalence among Swiss university students was reported of 7.8% (5.6% for alcohol and 2.5% for cannabis) [4]. Euphoric effects that occur after using drugs of abuse can be explained by the changes in the dopamine and serotonin levels in the brain, which might increase motivation [27], whereas frequent use has addictive potential and can generate anxiety, aggression, and paranoia [26]. Alcohol and cannabis use is related to increases in skipping class and lower grades due to interference with academic performance and assignments [28].

In our study, the prevalence of prescription drug use was 2.2%, almost beta-blockers (1.8%), and low use of methylphenidate (0.5%) and modafinil (0.1%). Drug consumption rates were higher for Romanian students than for French students. Studies among German students reported a similar prevalence, ranging between 0.26% and 2.0% [10]. Unlike the majority of European studies, which reported the preference of methylphenidates and modafinil by students, in our study, beta-blockers were the most used. Methylphenidate is prescribed to treat Attention-Deficit Hyperactivity Disorder (ADHD), modafinil is used to treat narcolepsy [29] and beta blockers, prescribed for cardiac arrhythmia, also have an anxiolytic effect [30]. Because of their extensive misuse by students, European governments have imposed strong restrictions on the prescription and delivery of methylphenidates and modafinil [31,32,33,34,35,36]. For example, dextroamphetamine and mixed amphetamine salts are prescribed to treat ADHD in the United States, whereas they are banned in Switzerland or Germany [34,35]. In France, these stimulants are subjected to a double prescription system. They are initially prescribed by psychiatrists and then the prescription should be confirmed by a general practitioner [30,32]. These modalities could limit methylphenidate and modafinil use but could lead to the research of other alternatives. Figures for the US are more worrying due to the misuse of stimulants prescribed for ADHD [36,37,38]. A meta-analysis of 21 US studies examining the prevalence of prescription drug misuse revealed that the past-year prevalence rates were 5% to 9% in high schools and 5% to 35% in colleges [36]. Methodologically, whereas the larger survey studies typically result in smaller rates of stimulant misuse, smaller, single-site studies with often face-to-face interviews, report higher risk. College practitioners and psychologists could assess for a learning disorder or attention deficit disorder to offer evidence-based medicine treatment. This would potentially reduce binge alcohol drinking and illicit substance use [39].

Smoking and binge drinking in recreational contexts were factors associated with drugs of abuse and soft enhancers for neuroenhancement. Our multivariate model strengthens the recent description in the US student population [40]. No significant association with eating disorders was detected for any of the three categories of substance. The association with methamphetamine has been recently demonstrated but despite a difference in our study, it was not significant, perhaps due to a lack of power [41].

Healthcare students are more frequent users of soft enhancers than other students. Healthcare students often experience emotional difficulties dealing with the challenges of their training [10], which can lead to burnout in some cases. Limiting the amount of knowledge and the psychological pressure because of competition during the first year may improve the use of psychostimulants in this population. Limiting sleep deprivation because of shifts may lessen psychostimulant use in healthcare students [35].

The primary reasons for using neuroenhancers reported by the university students were to stay awake (69.3%), to improve concentration (55.5%), to reduce stress (40.9%), and to improve memory (39.6%). The promotion of wakefulness was also reported by university students as one of the main reasons for neuroenhancement [36].

These findings indicate that challenges in the management of academic stress and workloads are behind most use of prescription and illicit drugs for neuroenhancement [4,22,31,42]. Students see cognitive enhancement as a way of coping with stressors and, therefore, increasing academic performance [37,43].

In our sample, 74.4% of users reported that neuroenhancement met their expectations and 40.5% reported an improvement in academic performance. However, it could be subjective effects. Several studies have called into question the cognitive performance benefits of drugs of abuse or prescription drugs in students [1,44,45]. A meta-analysis showed that expectations regarding the effectiveness of these drugs exceed their real effects [1]. This hypothesis was supported by a placebo-controlled study demonstrating that Modafinil affected the perceived change in physical performance and tiredness, but not cognitive performance in healthy adults [46]. Another double-blind placebo-controlled study reported that there were no significant differences in word recall tasks between sleep-deprived participants who received methylphenidate and those who received a placebo. However, significant differences were found between subjects who assumed they had received methylphenidate and those who assumed they had received a placebo [1]. Munro et al. cannot rule out the possibility that neuroenhancement prevented declines in academic performance, but conclude that students who engaged in neuroenhancement showed no increases in their academic performance and gained no detectable advantages over their peers [45]. Prescription drug consumption could affect neuroplasticity and may, therefore, result in deterioration of cognitive performance and even the personality of users [47].

In our study, many side effects were recorded, such as sleep disorders, palpitations, weariness, anxiety, loss of appetite, and aggressiveness. Another type of risk regarding the safety of cognitive neuroenhancers is addiction. A nationwide survey estimates that almost one in twenty misusers of prescription drugs meet the criteria for dependence or abuse [47]. Physiological changes in the brain caused by repeated use could lead to increased use of these drugs in more demanding academic environments [48]. Psychological addiction could also be generated because, with neuroenhancers, activities seem more interesting and rewarding [43], which might give the feeling that it is impossible to succeed without the drug use [49]. Furthermore, there is a real risk of delivering counterfeit medications [50] and a desire to use some smart drugs, but they did not use them, mainly due to the fear of side effects [51]. The medical safety and efficacy of prescription drugs varies with the substance used and side effects are not only pharmacological but also psychological and physiological. Repantis et al. [1] concluded that in the majority of trials, the drugs were well tolerated and, to some degree, improved memory, but there was no consistent evidence with repeated doses. A proper consumption would involve making enhancements available while managing their risks [52]. Ethical debates about neuroenhancement are between the issue of whether individuals have the right to use neuroenhancers and its potential social outcomes [52].

Our findings in this report have several limitations. The study did not include information from all the students at each of the three universities. It was a convenience sample, which could lead to a selection bias with an under- or overestimation of the neuroenhancement use. This bias could be limited by the anonymity of the questionnaire. Our French sample includes a bit more females than the origin population: Two-thirds of students are females in France, and almost three-quarters in Cluj-Napoca university. This convenience sample does not allow generalizing the results to other French and Romanian universities. However, we recruited a heterogeneous sample with respect to all major academic disciplines in the two French universities. The present study also has important strengths: The combination of similar studies at three different institutions in two countries, the use of specific questions regarding the broad range of neuroenhancement substances.

This study is one of the most detailed and consistent surveys shedding light on neuroenhancement among university students in Western and Eastern Europe. Specific questions concerning differences in neuroenhancement prevalence according to institution, substance category, and purpose of use were considered. Our results raise the alarm concerning potentially risky behavior among university students, particularly those in Romania. More measures are required to avoid neuroenhancement with drugs of abuse or prescription drugs and, thus, to limit other risky behaviors. Educational programs are also required to increase student awareness of the problems caused by neuroenhancement, and to change attitudes and beliefs to decrease the risk.

## 5. Conclusions

For the first time, different kinds of enhancers, motives, and factors associated with neuroenhancement have been highlighted among university students of Western and Eastern countries. Half of the students in the present study had used a substance at least once to improve academic performance, mostly coffee and vitamins. Neuroenhancement was strongly positively associated with frequent binge drinking and smoking behaviors. Despite the adverse effects experienced, most respondents reported that neuroenhancement fulfilled their expectations. Effective French-Romanian projects should be implemented to reduce the use of neuroenhancement and its negative consequences for health and social interactions. It would be relevant to continue with larger studies to narrow down the uses of neuroenhencers. Furthermore, prevention and screening materials tailored to Western and Eastern countries should be developed to help reduce substance use for neuroenhancement and its consequences among university students.

## 6. Strengths and Limitations

A wide range of use of products for neuroenhancement, prescription drugs, drugs of abuse, and soft enhancers were assessed. Motivations and characteristics of the users of neuroenhancement were identified among university students in France and Romania. It was a convenience sample, which could lead to a selection bias with an under or overestimation of the neuroenhancement use.

## Figures and Tables

**Figure 1 ijerph-18-03880-f001:**
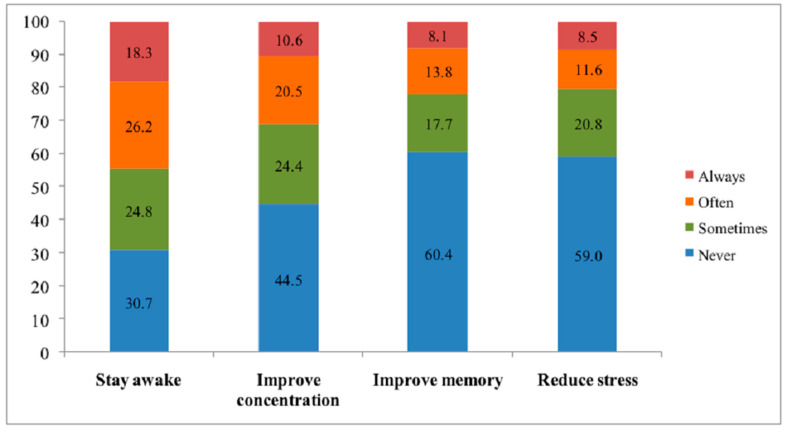
Motivations for neuroenhancement among university students from Cluj-Napoca, Rouen, and Opal Coast University (N = 611).

**Figure 2 ijerph-18-03880-f002:**
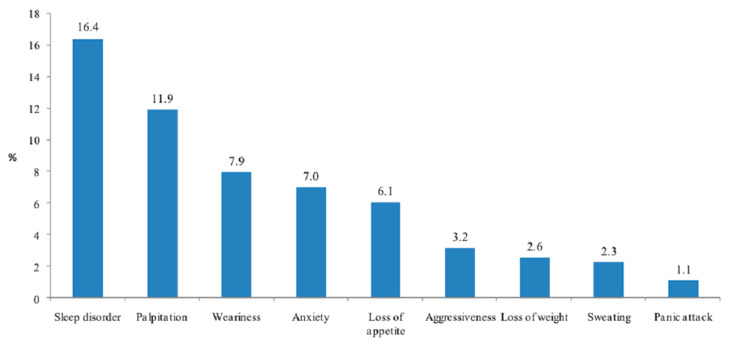
Adverse effects reported after neuroenhancer use by students at Cluj-Napoca, Rouen, and Opal Coast University.

**Figure 3 ijerph-18-03880-f003:**
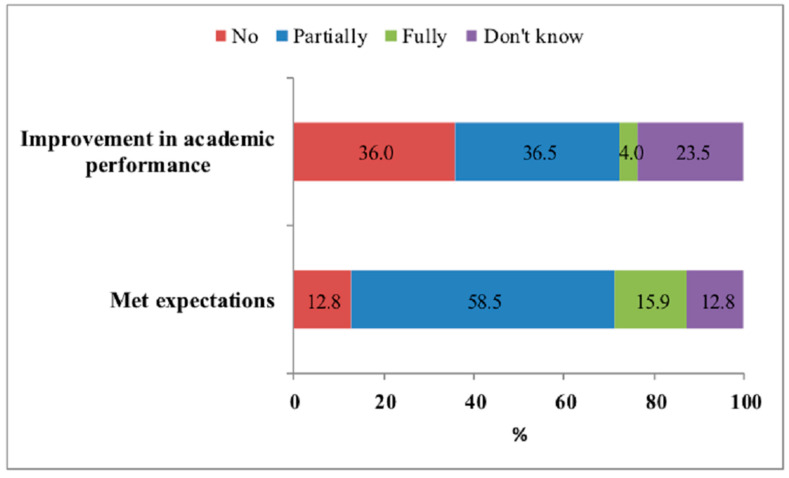
Satisfaction with neuroenhancer use for the university students from Cluj-Napoca, Rouen, and Opal Coast University.

**Table 1 ijerph-18-03880-t001:** Characteristics of university students from the three universities (*N* = 1110).

	Cluj-Napoca (*n* = 218)	Rouen (*n* = 534)	Opal Coast University (*n* = 358)
Female: *n* (%)	180 (82.5)	386 (72.3)	248 (70.7)
Mean age (SD)	21.4 (1.8)	20.1 (1.9)	19.7 (1.7)
Curriculum n (%)			
Mixed university group	0 (0.0)	326 (61.1)	78 (21.8)
Healthcare	218 (100)	208 (38.9)	280 (78.2)
Academic year of study n (%)			
1	37 (17.0)	231 (43.3)	259 (73.6)
2	70 (32.1)	75 (14.0)	54 (15.3)
3	26 (11.9)	110 (20.6)	24 (6.8)
>3	85 (39.0)	118 (22.1)	15 (4.3)

SD: standard deviation.

**Table 2 ijerph-18-03880-t002:** Prevalence of neuroenhancement, by substance category and study site, over the last 12 months (*N* = 1110).

	Cluj-Napoca (*n* = 218)	Rouen (*n* = 534)	Opal Coast University (*n* = 358)	Total (*N* = 1110)
Prescription drugs * (%)	6.4	1.3	0.8	2.2
Beta-blockers	6.0	1.2	1.1	1.8
Methylphenidate	0.9	0.2	0.3	0.5
Modafinil	0.0	0.0	0.3	0.1
Drugs of abuse * (%)	8.3	3.5	3.1	4.3
Alcohol	5.6	3.2	4.3	4.0
Cannabis	2.3	3.9	2.6	3.2
Amphetamines	2.7	0.8	0	0.8
Cocaine	0	0.7	0	0.4
Soft enhancers ** (%)	84.9	55.4	36.3	55.0
Coffee	80.7	43.4	25.9	45.3
Vitamins	53.2	23.7	15.3	26.9
Energy drinks	17.4	7.8	8.6	10.0
Caffeine tablets	8.2	1.3	2.0	2.9

**Table 3 ijerph-18-03880-t003:** Comparisons of the users of neuroenhancers of the three categories, according to sociodemographic characteristics, academic studies and risk behaviors among university students (*N* = 1110).

	Prescription Drugs			Drugs of Abuse			Soft Enhancers			Total
	Yes (*n* = 24)	No (*n* = 1086)	*p*-Value	Yes (*n* = 48)	No (*n* = 1062)	*p*-Value	Yes (*n* = 611)	No (*n* = 499)	*p*-Value	
Male (%)	65.2	74.0	0.34	58.3	74.5	0.01	76.0	71.0	0.06	73.8
Mean age (SD *)	21.9 (1.7)	20.2 (1.9)	0.89	21.0 (1.7)	20.2 (1.9)	0.64	20.5 (1.9)	19.8 (1.7)	<0.001	20.2 (1.9)
Student job (%)	16.7	13.3	0.63	22.9	12.9	0.05	13.0	13.9	0.64	13.4
Study grant-holder (%)	29.1	42.0	0.21	33.3	40.7	0.13	38.9	45.1	0.06	41.7
Curriculum (%)										
Mixed university group	8.3	37.0	0.004	35.4	36.5	0.88	31.3	42.1	<0.001	36.4
Healthcare	91.7	63.0		64.6	63.5		68.7	57.9		63.6
Academic year of study (%)										
1	12.5	48.5		31.2	48.5		39.7	57.6		47.7
2	20.8	18.0	<0.001	20.9	17.9	0.12	20.8	14.6	<0.001	18.1
3	16.7	14.4		20.9	14.2		14.9	14.0		14.5
>3	50.0	19.1		27.0	19.4		24.6	13.8		19.7
Mean stress (SD *)	20.6 (6.1)	16.7 (7.4)	0.43	19.7 (7.8)	16.7 (7.3)	0.85	17.6 (7.3)	15.9 (7.5)	0.96	16.8 (7.4)
Tobacco smoking (%)	25.0	21.4	0.67	56.3	19.9	<0.001	26.9	14.8	<0.001	21.5
Binge drinking (%) Never Occasional Frequent	50.0 41.7 8.3	44.1 48.8 7.1	0.78	31.3 45.8 22.9	44.9 48.8 6.3	<0.001	42.6 49.1 8.3	46.4 48.1 5.5	0.17^4^	44.3 48.6 7.1
Eating disorders (%)	33.3	25.1	0.36	33.3	24.9	0.19	26.7	23.5	0.21	25.3

SD: Standard Deviation.

**Table 4 ijerph-18-03880-t004:** Logistic regression analysis of the factors associated with neuroenhancement among university students.

KERRYPNX	Drugs of Abuse		Soft Enhancers	
	AOR 95% CI	*p*-Value	AOR 95% CI	*p*-Value
Model 1				
Male	2.23 (1.21–4.11)	0.01	0.89 (0.66–1.18)	0.42
Student job	2.42 (1.13–5.17)	0.001		
Study grant-holder	0.68 (0.35–1.33)	0.27	0.98 (0.75–1.28)	0.90
Curriculum				
Mixed university group			1 (Ref)	
Healthcare			1.38 (1.01–1.87)	0.04
Academic year of study				
1				
2	1 (Ref)		1 (Ref)	
3	1.23 (0.51–2.96)	0.64	1.22 (0.83–1.79)	0.30
>3	1.79 (0.74–4.32)	0.19	0.93 (0.60–1.45)	0.77
	1.37 (0.58–3.24)	0.46	0.90 (0.75–1.28)	0.71
Model 2				
Tobacco smoking	5.50 (2.98–10.14)	<0.001	2.71 (1.94–3.80)	<0.001
Model 3				
Binge drinking				
Never	1 (Ref)	-	1 (Ref)	
Occasional	1.55 (0.76–3.19)	0.23	1.69 (1.26–5.92)	<0.001
Frequent	6.49 (2.53–16.60)	<0.001	3.34 (1.88–4.13)	0.001

Model 1 included sociodemographic variables (sex, age, university, student job, grant-holder status, curriculum, academic year). Model 2 was defined as model 1 + smoking and Model 3 as model 1 + binge drinking.

## Data Availability

All available data can be obtained by contacting the corresponding author.

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
