# Peer review of "Neuroenhancement in French and Romanian University Students, Motivations and Associated Factors"

_ijerph, 2021, doi:10.3390/ijerph18083880_

Round 1
Reviewer 1 Report
Performance is one word, no hyphenation. Line 48 period and comma (please fix punctuation all through-out your article). Do not hyphenate University. Same for conducted. Undergraduate. Hypothesis. The PSS is not a diagnostic instrument, therefore I am not sure why you used it? Performance is not hyphenated. There is no discussion about binge eating/or eating disorders in your summary. I would recommend revising your grammar, no hyphenated words, and capitalize your subheadings. Revised the summary to introduce new ideas and the significance to academia and college students. Should there be college psychologists that can assess for a learning disorder or attention deficit disorder to offer evidence based medicine treatment. Probably yes. This would reduce potentially binge alcohol drinking and illicit substance use.
Author Response
Performance is one word, no hyphenation.
It was an error of the pdf document (in the word document there is no hyphenation). The new pdf is corrected.
Line 48 period and comma (please fix punctuation all through-out your article).
The punctuation has been completely revised throughout the article. We corrected spelling/grammatical errors
Do not hyphenate University. Same for conducted. Undergraduate. Hypothesis.
It was an error of the pdf document (in the word document there is no hyphenation). The new pdf is corrected.
The PSS is not a diagnostic instrument, therefore I am not sure why you used it?
Indeed, this scale give not a diagnosis but a measure of the relative stress level with linear relationship between the core and the stress: higher scores correspond to higher levels of perceived stress. Our results show that substance use is not associated with stress
Performance is not hyphenated.
There is no discussion about binge eating/or eating disorders in your summary.
We added a sentence in the abstract :’No significant association with eating disorders were detected for any of the three categories of substance’
I would recommend revising your grammar, no hyphenated words, and capitalize your subheadings.
We corrected spelling/grammatical errors
Revised the summary to introduce new ideas and the significance to academia and college students.
We add this sentence: “Educational programs in university seems to be required to increase student awareness of the problems caused by neuroenhancement, and to change attitudes and beliefs to decrease the risk”.
Should there be college psychologists that can assess for a learning disorder or attention deficit disorder to offer evidence based medicine treatment. Probably yes. This would reduce potentially binge alcohol drinking and illicit substance use.
As suggested we added the sentence “College practioner and psychologists could assess for a learning disorder or attention deficit disorder to offer evidence based medicine treatment This would reduce potentially binge alcohol drinking and illicit substance use”
Reviewer 2 Report
I read carefully the paper entitled: ”Neuroenhancement in French and Romanian university students, motivations and associated factors”.
The paper is presented at a good scientific level.
We really appreciate that ”This survey is the first to provide information about the use of a large range of products (prescriptions drugs, drug of abuse and soft enhancers), reasons for neuroenhancement and about the factors associated with neuroenhancement for university students at French and Romanian universities...”.
Authors' arguments, methods used, data, etc. are in line with the requirements of scientific research.
1. However, the Abstract must be written as a sum of sentences, without giving the exact structure of the paper - ”Objectives: …”, ”Participants: ….”, ”Results: …..”, ”Conclusion: …”.
2. I appreciate that a part of the "Literature Review" was required, with reference to several papers on similar topics.
3. Line 306: source [xxxxx] is omitted.
4. Lines 47, 308, 313: punctuation problems.
Small errors can be easily fixed. If they read carefully the text of the paper, the authors will detect for themselves the negligences...
5. The conclusions of the paper should be extended by a few sentences.
6. But he must apply the rules of the Journal.
7. It would be good to expand the bibliography with some articles from prestigious scientific journals (WoS) published in 2019-2020-2021.
8. All in all, eventually, it may be seen by an English teacher (native).
Author Response
I read carefully the paper entitled: ”Neuroenhancement in French and Romanian university students, motivations and associated factors”.
The paper is presented at a good scientific level.
We really appreciate that ”This survey is the first to provide information about the use of a large range of products (prescriptions drugs, drug of abuse and soft enhancers), reasons for neuroenhancement and about the factors associated with neuroenhancement for university students at French and Romanian universities...”.
Authors' arguments, methods used, data, etc. are in line with the requirements of scientific research.
1.However, the Abstract must be written as a sum of sentences, without giving the exact structure of the paper - ”Objectives: …”, ”Participants: ….”, ”Results: …..”, ”Conclusion: …”.
We delete the heading of the abstract
- I appreciate that a part of the "Literature Review" was required, with reference to several papers on similar topics.
- Line 306: source [xxxxx] is omitted.
- Lines 47, 308, 313: punctuation problems.
Small errors can be easily fixed. If they read carefully the text of the paper, the authors will detect for themselves the negligences...
We corrected spelling/grammatical errors
- The conclusions of the paper should be extended by a few sentences.
As suggested by the reviewer we add the sentences: “It would be relevant to continue with large studies to narrow down the uses of neuroenhencer. Furthermore prevention and screening materials tailored to Western and Eastern countries should be developed to help reduce substance use for neuroenhancement and its consequences among university students.”
- But he must apply the rules of the Journal.
We thank the reviewer and apply the rules.
- It would be good to expand the bibliography with some articles from prestigious scientific journals (WoS) published in 2019-2020-2021.
Recently references added: METTRE LES NUMERO
- All in all, eventually, it may be seen by an English teacher (native).
English has been revised by a native speaker
Reviewer 3 Report
This paper aimed to determine the use of neuroenhancer, motivations and the factors associated with their use in French and Romanian university students.
Unfortunately, the results obtained, the factors, were within expectations, but this study will have important consequences for protecting the health of university students. Additionally, clarifying them numerically would be important to take epidemiological studies to the next step.
However I considered there are various opinions about the importance of the results obtained, it seems that the statistical method used in this study is correct.Therefore, I have no reason to object to the acceptance of this paper.It would be better to emphasize the novelty of the results in the conclusion section. If possible, reconsider.
Author Response
This paper aimed to determine the use of neuroenhancer, motivations and the factors associated with their use in French and Romanian university students.
Unfortunately, the results obtained, the factors, were within expectations, but this study will have important consequences for protecting the health of university students. Additionally, clarifying them numerically would be important to take epidemiological studies to the next step.
However I considered there are various opinions about the importance of the results obtained, it seems that the statistical method used in this study is correct.
We thank the reviewer for his comments
Therefore, I have no reason to object to the acceptance of this paper.It would be better to emphasize the novelty of the results in the conclusion section. If possible, reconsider.
As suggested by the reviewer, we added the sentence in tbe conclusion section :”For the first time, all kinds of enhancers, motives and factors associated with neuroenhancement has been highlighted among university students of Western and Eastern countries.”
Reviewer 4 Report
The present study is of great interest and highlights important issues regarding both the extent and reasons for students' use of all kinds of 'enhancers'. The sampling is convenient, therefore non-probabilistic, but the number of participants in the study reveals important differences between Western and Eastern European students. The data from this study can be used, among other things, when designing prevention strategies, especially in Eastern European countries.
The manuscript was prepared with great care. Sample selection is described correctly, applied tools and research procedure are adequate to the research. The procedure of statistical data analysis has been described clearly. Selection of statistical tools adequate to undertaken analysis.
For better organization and in accordance with methodological rigor, the aim of the study should be presented in a separate section (it is at the end of the "Introduction" section). The main research assumptions can also be presented there.
Also, I noticed two minor errors: on line 218, a name is inserted in square brackets (which should not be there); and on line 306, there is an empty bracket with a missing reference (presumably to item 48 from References).
The manuscript (in the "Discussion" section) highlights one very important problem: the presence of differences in results from studies conducted in a broader and narrower scope, where the latter report a higher risk of stimulant abuse. This problem can - and should - be verified in well-designed and reliably conducted comparative empirical studies. Perhaps the use of machine learning and artificial intelligence methods would help to establish the suggested trends. In my opinion, this is a very important conclusion.
Author Response
The present study is of great interest and highlights important issues regarding both the extent and reasons for students' use of all kinds of 'enhancers'. The sampling is convenient, therefore non-probabilistic, but the number of participants in the study reveals important differences between Western and Eastern European students. The data from this study can be used, among other things, when designing prevention strategies, especially in Eastern European countries.T he manuscript was prepared with great care. Sample selection is described correctly, applied tools and research procedure are adequate to the research. The procedure of statistical data analysis has been described clearly. Selection of statistical tools adequate to undertaken analysis.
We thank the reviewer for his comments
For better organization and in accordance with methodological rigor, the aim of the study should be presented in a separate section (it is at the end of the "Introduction" section). The main research assumptions can also be presented there.
The rules for the author are for the introduction :”Finally, briefly mention the main aim of the work …” then we put the objective at the end of the introduction
Also, I noticed two minor errors: on line 218, a name is inserted in square brackets (which should not be there); and on line 306, there is an empty bracket with a missing reference (presumably to item 48 from References).
We thank the reviewer and corrected the error.
The manuscript (in the "Discussion" section) highlights one very important problem: the presence of differences in results from studies conducted in a broader and narrower scope, where the latter report a higher risk of stimulant abuse. This problem can - and should - be verified in well-designed and reliably conducted comparative empirical studies. Perhaps the use of machine learning and artificial intelligence methods would help to establish the suggested trends. In my opinion, this is a very important conclusion.
We thank the reviewer for this comment. Artificial intelligence could be used from big data. In France there is the national data of drug consumption reimbursed by the health insurance but the student status and the consumption of non-drug substances are not in this database. Maybe in other countries it would be possible?
A sentence was added in the discussion: “It would be relevant to continue with large studies to narrow down the uses of neuroenhencer”